# Improved Electrochemical Performance of Li-Rich Cathode Materials via Spinel Li_2_MoO_4_ Coating

**DOI:** 10.3390/ma16165655

**Published:** 2023-08-17

**Authors:** Shuhao Zhang, Yun Ye, Zhaoxiong Chen, Qinghao Lai, Tie Liu, Qiang Wang, Shuang Yuan

**Affiliations:** 1School of Metallurgy, Northeastern University, Shenyang 110819, China; 2Key Laboratory for Ecological Metallurgy of Multimetallic Mineral, Ministry of Education, Northeastern University, Shenyang 110819, China; 3Key Laboratory of Electromagnetic Processing of Materials, Ministry of Education, Northeastern University, Shenyang 110819, China

**Keywords:** LIBs, Li-rich manganese-based cathode material, Li_2_MoO_4_ coating, spinel phase, electrochemical performance

## Abstract

Li-rich manganese-based cathode materials (LRMs) are considered one of the most promising cathode materials for the next generation of lithium-ion batteries (LIBs) because of their high energy density. However, there are problems such as a capacity decay, poor rate performance, and continuous voltage drop, which seriously limit their large-scale commercial applications. In this work, Li_1.2_Mn_0.54_Co_0.13_Ni_0.13_O_2_ coated with Li_2_MoO_4_ with a unique spinel structure was prepared with the wet chemistry method and the subsequent calcination process. The Li_2_MoO_4_ coating layer with a spinel structure could provide a 3D Li^+^ transport channel, which is beneficial for improving rate performance, while protecting LRMs from electrolyte corrosion, suppressing interface side reactions, and improving cycling stability. The capacity retention rate of LRMs coated with 3 wt% Li_2_MoO_4_ increased from 69.25% to 81.85% after 100 cycles at 1 C, and the voltage attenuation decreased from 7.06 to 4.98 mV per cycle. The lower R_ct_ also exhibited an improved rate performance. The results indicate that the Li_2_MoO_4_ coating effectively improves the cyclic stability and electrochemical performance of LRMs.

## 1. Introduction

LIBs as a kind of clean and renewable electrochemical energy storage equipment have been widely applied. However, the current energy density of commercial LIBs is insufficient to meet the growing demand, and the development of high-capacity electrode materials is urgent [1,2]. Li-rich manganese-based cathode materials (LRMs) are considered to be one of the most promising cathode materials for the next generation of LIBs due to their high specific capacity (>250 mAh/g), low cost, and excellent energy density (>1000 Wh/kg) [3,4].

LRMs are cathode materials with a layered structure, in which the Li layer and transition metal (TM) layer are arranged alternately, and the general formula is xLi_2_MnO_3_·(1−x)LiMO_2_ (M = Ni, Co, and Mn). This expression emphasizes that LRMs are two-phase composite structures composed of rhombohedral LiMO_2_ (R-3m) and monoclinic Li_2_MnO_3_ (C2/m) [5]. Based on the oxidation–reduction of TM and the additional oxygen redox, more Li can be extracted to provide an ultra-high capacity [6,7]. Specifically, taking the first charge process as an example, when the voltage is below 4.5 V, TM ions are oxidized to a higher valence and Li^+^ is extracted from the LiMO_2_ component. When the voltage is above 4.5 V, oxygen ions participate in the oxidation process, and Li^+^ is extracted from the Li_2_MnO_3_ component, providing additional capacity [1]. However, the practical application of LRMs is impeded by a severe voltage and capacity decay, mainly due to the continuous migration of TM and the resulting structural evolution from layered to spinel or rock salt phases [8,9].

TM migration or structural evolution begins on the surface of LRMs during the cyclic process [10,11,12]. The continuous extraction of Li at a high operating voltage and severe interfacial reaction between the electrode and electrolyte lead to the migration of surface TM towards the Li layer [10], resulting in the structural evolution of layered to spinel or rock salt phases. And this phase transition will gradually propagate towards the interior of LRMs during the cycling process [13], so the surface protection of LRMs is very crucial. Surface coating is an effective surface protection method. Different coating materials can play different roles, such as reducing electrolyte side reactions, stabilizing the electrode–electrolyte interface, improving the thermal stability of electrode materials, inhibiting gas generation, alleviating structural degradation and crack generation under a high operating voltage, improving ion transport, and improving electronic conductivity [14]. So far, many materials, such as oxides Al_2_O_3_ [15], ZrO_2_ [16], fluoride AlF_3_ [17], LiF [18], and metal phosphate LiFePO_4_ [19], have been used as surface coatings to improve the cyclic stability of LRMs. However, although these efforts have made significant contributions to improving the cycle life, most coatings have a poor compatibility with LRMs, which is not conducive to the uniform protection of LRMs and maintaining overall integrity with LRMs over long cycles [20]. Therefore, it is highly desirable to introduce lithium-reactive coating materials to enhance the compatibility between the coating and the material body.

In this work, we coated Li_2_MoO_4_ with a unique spinel structure on the surface of Li_1.2_Mn_0.54_Co_0.13_Ni_0.13_O_2_. Through the wet chemistry method and the subsequent calcination process, Li_2_MoO_4_ can be evenly coated on the surface of LRMs. In addition, when molybdate reacts with the cathode material at a high temperature, it can form a spinel phase on the surface of the material by removing part of Li. The spinel structure has 3D Li^+^ channels, which are conducive to Li^+^ transport. At the same time, the Li_2_MoO_4_ coating, as a good lithium ion conductor, can provide better protection, preventing LRMs from electrolyte corrosion, thus improving the structural stability and cycle life.

## 2. Materials and Methods

### 2.1. Material Synthesis

The pristine Li_1.2_Mn_0.54_Co_0.13_Ni_0.13_O_2_ was synthesized with a high-temperature solid-state method. In detail, the carbonate precursor (Mn_0.675_Co_0.1625_Ni_0.1625_CO_3_) from HaiAnZhiChuan Battery Materials Technology Co., Ltd. (Nantong, China) was uniformly ground and mixed with Li_2_CO_3_ (an excess of 5% Li_2_CO_3_ was added to compensate for the Li loss during the elevated calcination step) in an agate mortar. Subsequently, the powder mixture was sintered at 500 °C for 6 h and then 850 °C for 12 h to produce the target Li_1.2_Mn_0.54_Co_0.13_Ni_0.13_O_2_ (Pristine).

The prepared Li-rich materials were mixed with a certain amount of ammonium molybdate. To avoid agglomeration, the mixed materials were dissolved in absolute ethanol after grinding and stirred for 3 h. After evaporation in a water bath at 80 °C, the desiccative powder was calcined at 720 °C for 4 h to obtain the Li_2_MoO_4_@Li_1.2_Mn_0.54_Ni_0.13_Co_0.13_O_2_ sample. In order to explore the optimal coating amount, 1, 3, and 5 wt% ammonium molybdate were mixed with Li-rich materials, respectively, and the corresponding samples were called 1 wt%, 3 wt%, and 5 wt%.

### 2.2. Material Characterization

X-ray diffraction patterns of samples for the crystal structure were obtained with an X-ray powder diffractometer (XRD, D8 Advance, German Bruker AXS Co., Ltd., Karlsruhe, Germany, Cu Kα radiation, λ = 0.15406 nm), and XRD data were obtained in an angular range of 10–80° with a scanning speed of 1°/min. The test results were refined using GSAS [21]. Scanning electron microscopy (SEM, Thermo Scientific Apreo 2c, Thermo Fisher Scientific, Waltham, MA, USA) was applied to observe the morphology of the material. Transmission electron microscopy (TEM, JEM2200FS, JEOL, Tokyo, Japan) was used to detect the structure and microstructure of the samples. An inductively coupled plasma emission spectrometer (ICP-OES, Agilent 5110, Agilent Technology Co., Ltd., Santa Clara, CA, USA) was used to determine the composition of the elements. An X-ray photoelectron spectroscopy analyzer (XPS, Thermo ESCALAB 250XI, Thermo Fisher Scientific, Waltham, MA, USA) was used to measure element species and valence state information on the sample surface.

### 2.3. Electrochemical Characterization

The active cathode material, conductive agent acetylene black, and binder PVDF were weighed in a mass ratio of 8:1:1 and mixed evenly in a N-methyl-2-pyrrolidone (NMP) solvent to prepare the cathode slurry. The prepared slurry was uniformly coated on aluminum foil, and then dried in an oven at 60 °C for 12 h. The dry electrode was cut into 12 mm wafers with a slicer, and the mass of the electrode was weighed after pressing. The load of active cathode material was about 1.5 mg/cm^2^. The button cell was assembled in a glove box filled with Ar (H_2_O and O_2_ contents were lower than 0.1 ppm). An LB-111 high-voltage electrolyte from DoDochem Company (Suzhou, China) was used as the electrolyte.

The electrochemical performance was tested on LAND, and carried out in an incubator at 30 °C. The battery was tested in the voltage range of 2.0–4.8 V. For the cycling performance, the battery was first cycled for 3 times at 0.1 C (1 C = 250 mA/g), and then cycled to 100 times at 1 C. For the rate performance, the battery was cycled for five times at 0.1, 0.2, 0.5, 1, 2, and 5 C, respectively, and finally at 0.1 C. The cyclic voltammetry (CV) curve and electrochemical impedance spectroscopy (EIS) were performed on a VSP electrochemical workstation. The CV scanning rate was 0.1 mV/s and the voltage range was 2.0–4.8 V. The EIS frequency range was 0.01–100 Hz, and the AC amplitude was 5 mV.

## 3. Results and Discussion

### 3.1. Material Characterizations

Previous research on Li_2_MoO_4_ coating basically covered the surface of high-nickel ternary cathode materials with Li_2_MoO_4_, and formed Li_2_MoO_4_ coating by using residual lithium on the surface of high-nickel materials, while eliminating the adverse effects caused by surface residual lithium [22,23,24,25]. In this work, Li_2_MoO_4_ was coated on the surface of Li-rich materials. Unlike high-nickel materials, Li-rich materials have no residual lithium on the surface, so the Li forming Li_2_MoO_4_ coating comes from the surface of Li-rich materials, as shown in Formulas (1) and (2) (Li_1.2_Mn_0.54_Co_0.13_Ni_0.13_O_2_ is abbreviated as Li_1.2_TM_0.8_O_2_).
Li_1.2_TM_0.8_O_2_→0.4LiTM_2_O_4_ + 0.4Li_2_O(1)
Li_2_O + MoO_3_→Li_2_MoO_4_(2)

At a high temperature, molten molybdate leaches Li_2_O from the surface of Li-rich materials, and then anneals the remaining crystals at a high temperature to eliminate Li/O vacancies and form spinel LiTM_2_O_4_ on the surface [26]. The leached Li_2_O and molybdate (represented by MoO_3_ in the formula) form Li_2_MoO_4_ coating. The difference is that this Li_2_MoO_4_ coating has a unique spinel structure, which will be specifically explained in the following structural analysis.

The element content of the prepared materials was analyzed with an inductively coupled plasma optical emission spectrometer (ICP-OES). The results are listed in Appendix A. The experimental results are in good agreement with the design values.

Figure 1 shows the X-ray powder diffraction (XRD) patterns of the samples and their Rietveld refinement results. Figure 1a shows that the layered α-NaFeO_2_ structure with an R-3m space group exists in the four samples, and the characteristic peaks in the range of 20–25° also indicate the existence of the Li_2_MnO_3_ phase with a C2/m space group in the samples. The clear split of the (006)/(102) and (018)/(110) peaks implies a highly ordered layered structure of the samples [27]. In addition, other peaks can be clearly observed near 18, 30, and 35°, and the peak intensity increases with the increase in the Li_2_MoO_4_ coating amount. After the analysis and comparison, it can be determined that the weak peak is the spinel structure of Li_2_MoO_4_ (Appendix A [26]).

The XRD patterns of the four samples were refined to a two-phase structure model consisting of rhombohedral R-3m and monoclinic C2/m phases, and the results are shown in Figure 1b–e. According to reports, the diffraction peak intensity ratio between the planes (003) and (104) can be used to estimate the degree of Li/Ni mixing. The smaller the ratio, the greater the degree of Li/Ni mixing [28]. As shown in Table 1, the ratios I(003)/I(104) of all samples are greater than 1.2, indicating a lower Li/Ni miscibility [29]. Compared to the Pristine sample, the ratio I(003)/I(104) of the sample coated with Li_2_MoO_4_ decreased slightly, possibly due to the intensification of Li/Ni mixing during high-temperature calcination during the preparation process. Among them, the ratio I(003)/I(104) of the 3 wt% sample is closest to the Pristine sample, and the content of Ni^2+^ in the Li layer in the refined results of Table 1 also corresponds to the ratio I(003)/I(104). Additionally, the spacing between layers of the sample coated with Li_2_MoO_4_ increased slightly, possibly due to the partial doping of Mo in the TM layer on the surface. The subsequent X-ray photoelectron spectroscopy (XPS) results provide evidence for this result. However, the ratio between the lattice parameters c and a of the four samples is greater than 4.99, indicating that the samples have a fine-layered structure [30]. At the same time, the content of Li_2_MoO_4_ in the coated samples was refined and the results are shown in Table 1. Except for the low peak intensity of Li_2_MoO_4_ in the 1 wt% sample, which cannot obtain the content of Li_2_MoO_4_, the content of Li_2_MoO_4_ in the 3 wt% and 5 wt% samples is in good agreement with the design values and ICP results.

The morphological characteristics of the prepared samples were investigated with a scanning electron microscope (SEM), and the obtained images are shown in Appendix A. The morphology of all the sample particles is spherical and the diameter of the secondary particles ranges from 10 to 15 μm. Primary particles can be clearly seen on the surface of the Pristine sample particles, but the surface of the coated Li-rich material becomes smooth.

A transmission electron microscope (TEM) and fast Fourier transform (FFT) were used to display the crystal structure near the surface areas of the 3 wt% sample. The Li_2_MoO_4_ coating layers on the surface of the material can be clearly observed in Figure 2a. Figure 2b shows clear lattice stripes of the layered structure. Combining with the FFT image (Figure 2e) of the corresponding position in Figure 2d, it can be determined that the spacing of lattice stripes is 0.474 nm, which is very consistent with the plane (003) of the R-3m space group of the Li-rich layered structure. Through the analysis of the FFT image of the blue region in Figure 2c, it can be found that a spinel structure coating is formed on the surface of the 3 wt% sample. The thickness of the coating can reach over 20 nm due to not only the unique spinel structure of Li_2_MoO_4_ formed with coating but also the spinel phase formed with the detachment of some Li from the material surface during the formation of Li_2_MoO_4_ coating (Formulas (1) and (2)). The elemental distribution on the surface of the 3 wt% sample was analyzed with EDS. As shown in Figure 2f–k, Mn, Co, Ni, O, and Mo elements are distributed uniformly, indicating that Li_2_MoO_4_ is successfully coated on the surface of Li-rich materials.

The surface elemental composition and chemical state of the Pristine and 3 wt% samples were analyzed with X-ray photoelectron spectroscopy (XPS) (Figure 3 and Appendix A). Figure 3a,b shows the XPS full spectrum of the Pristine and 3 wt% samples, respectively, indicating the presence of Mo on the surface of the materials. Figure 3c shows the XPS spectra of Mo in the 3 wt% sample, with peaks near 232 eV and 235 eV corresponding to Mo 3d_5/2_ and Mo 3d_3/2_, respectively. Mo^6+^ indicates the presence of Li_2_MoO_4_ coating on the surface of the 3 wt% sample, while the presence of Mo^4+^ indicates that some Mo^6+^ is doped into the lattice to convert into Mo^4+^ to maintain electrical neutrality [24]. Appendix A shows the XPS spectra of Mn, Co, and Ni in the Pristine and 3 wt% samples, respectively. There is no significant change in the valence states of Mn, Co, and Ni, indicating that Li_2_MoO_4_ coating will not have a significant impact on the original system state of Li-rich materials.

### 3.2. Electrochemical Performances

Figure 4a shows the initial charge–discharge curves of four samples at 0.1 C in the voltage range of 2.0–4.8 V. The Pristine, 1 wt%, 3 wt%, and 5 wt% samples release 259.65, 250.29, 257.31, and 248.74 mAh/g, respectively, corresponding to Coulombic efficiencies of 56.66, 76.92, 77.79, and 75.16%. The first charging curve has two charging platforms, which is a typical feature of Li-rich materials. The inclined plateau below 4.5 V belongs to the oxidation reaction of Ni^2+^/Ni^4+^ and Co^3+^/Co^4+^, while the long plateau near 4.5 V is related to the oxygen activation of Li_2_MnO_3_. At a high voltage, the Li_2_MnO_3_ phase releases Li and O (main forms are Li_2_O and O_2_), and O_2_ will escape from the material, causing an irreversible capacity loss. Therefore, the first cycle discharge capacity of Li-rich materials is smaller than the charge capacity, indicating a low initial Coulombic efficiency [31,32]. Compared to the Pristine sample, the platform of the sample coated with Li_2_MoO_4_ becomes shorter near 4.5 V. This is because the formation of the Li_2_MoO_4_ coating layer removes some Li from the material (Formula (1)) and pre-activates the Li_2_MnO_3_ component [33,34]. Therefore, the sample coated with Li2MoO_4_ has a higher initial Coulombic efficiency. The cycling performance of four samples at 1 C is shown in Appendix A. For comparison purposes, only the cycling performance of the Pristine and 3 wt% samples is shown in Figure 4b. The results showed that the Li_2_MoO_4_ coating improved the cycling performance of the material, with only a 69.25% capacity retention rate of the Pristine sample after 100 cycles, while the 3 wt% sample still had an 81.85% capacity retention rate after 100 cycles. This is attributed to the fact that the Li_2_MoO_4_ coating effectively protects the internal Li-rich materials from electrolyte corrosion, inhibits interface side reactions, and improves the stability and cycling performance of lithium-ion batteries.

Figure 4c shows the rate performance of the samples. The discharge capacity of all four samples decreases with the increase in current density, with the 3 wt% sample exhibiting the best rate performance. The spinel has a 3D Li^+^ channel compared to the 2D Li^+^ channel of the Li-rich materials, which favors an improvement in the rate capability [35]. Meanwhile, Li_2_MoO_4_ belongs to the fast-ion conductor, which is conducive to Li^+^ transport. Therefore, the material coated with Li_2_MoO_4_ has a higher rate performance [36].

In addition, according to Figure 4d,e, the Pristine sample exhibits a faster voltage and capacity decay, which is also one of the main problems of Li-rich materials. Figure 4f shows that the average voltage decay of the 3 wt% sample (a voltage drop of 4.98 mV per cycle, 100 cycles) is less than that of the Pristine sample (a voltage drop of 7.06 mV per cycle, 100 cycles) at a high cutoff voltage (4.8 V). The increase in capacity and discharge voltage retention of the 3 wt% sample is due to the effective reduction in direct contact between the electrode and the electrolyte with the Li_2_MoO_4_ coating, which inhibits the erosion of Li-rich materials during discharge, prevents the oxidation of the electrolyte on the surface of the cathode material during charging, and improves its structural stability.

The kinetic properties of Li-rich materials before and after Li_2_MoO_4_ coating were studied with electrochemical impedance spectroscopy (EIS). Before analyzing the EIS results, a Kramers–Kronig residual analysis was performed for data validation. The results showed that the EIS data met the Kramers–Kronig relationship and could be used to study the kinetic properties of the prepared samples (Appendix A [37,38]). As shown in Figure 5a, the EIS curve consists of a semicircle in the high-frequency region and a slanted line in the low-frequency region, where the diameter of the semicircle represents the charge transfer resistance (R_ct_) and the slanted line represents the Warburg impedance. The equivalent circuit and the impedance parameters fitted to four samples are shown in Figure 5b. As expected, R_ct_ decreased after coating with Li_2_MoO_4_, which was attributed to the spinel structure of the Li_2_MoO_4_ coating that is capable of providing 3D channels for Li^+^ diffusion, which is favorable for Li^+^ diffusion.

Figure 5c shows the CV curve of the 3 wt% sample between 2.0 and 4.8 V. During the first charge–discharge cycle, two oxidation peaks near 4.1 and 4.7 V are clearly observed, corresponding to the oxidation reactions of Ni^2+^/Ni^4+^ and Co^3+^/Ni^4+^, as well as the activation of the Li_2_MnO_3_ phase, respectively. The activation of the Li_2_MnO_3_ phase is accompanied by the irreversible release of O_2_, which only occurs during the first charge–discharge cycle. Therefore, in the subsequent cycles, the oxidation peak near 4.7 V disappears [39], which is also the reason for the low initial Coulombic efficiency of Li-rich materials.

## 4. Conclusions

In summary, Li_1.2_Mn_0.54_Co_0.13_Ni_0.13_O_2_ coated with Li_2_MoO_4_ was successfully prepared. XRD results indicate that the samples coated with Li_2_MoO_4_ exhibited a good layered structure, while the Li_2_MoO_4_ coating exhibited a unique spinel structure. The SEM and TEM results indicate that Li_2_MoO_4_ was perfectly coated on the surface of the material. To determine the optimal amount of coating, a series of Li-rich materials coated with Li_2_MoO_4_ were prepared, among which the 3 wt% Li_2_MoO_4_ sample exhibited the best electrochemical performance. Compared to the Pristine sample, the initial Coulomb efficiency of the sample coated with Li_2_MoO_4_ greatly improved, from 56.66% to over 75%, effectively alleviating the problem of a low initial Coulomb efficiency in Li-rich materials. After 100 cycles at 1 C, the Pristine sample only had a capacity retention rate of 69.25%, while the sample coated with 3 wt% Li_2_MoO_4_ had a capacity retention rate of 81.85%, greatly improving cycling performance. Meanwhile, the rate performance curve also indicates that the sample coated with 3 wt% Li_2_MoO_4_ had a higher capacity at a high rate. Furthermore, the voltage attenuation of Li-rich materials coated with Li_2_MoO_4_ was also suppressed. Compared to the Pristine sample, the voltage attenuation of the sample coated with 3 wt% Li_2_MoO_4_ slowed by an average of 2.08 mV per 100 cycles. The EIS results showed that the R_ct_ of the sample coated with Li_2_MoO_4_ decreased significantly and the diffusion of Li^+^ was enhanced. This work indicates that the Li_2_MoO_4_ coating can enhance the cyclic stability of Li-rich materials and improve their electrochemical performance.

## Figures and Tables

**Figure 1 materials-16-05655-f001:**
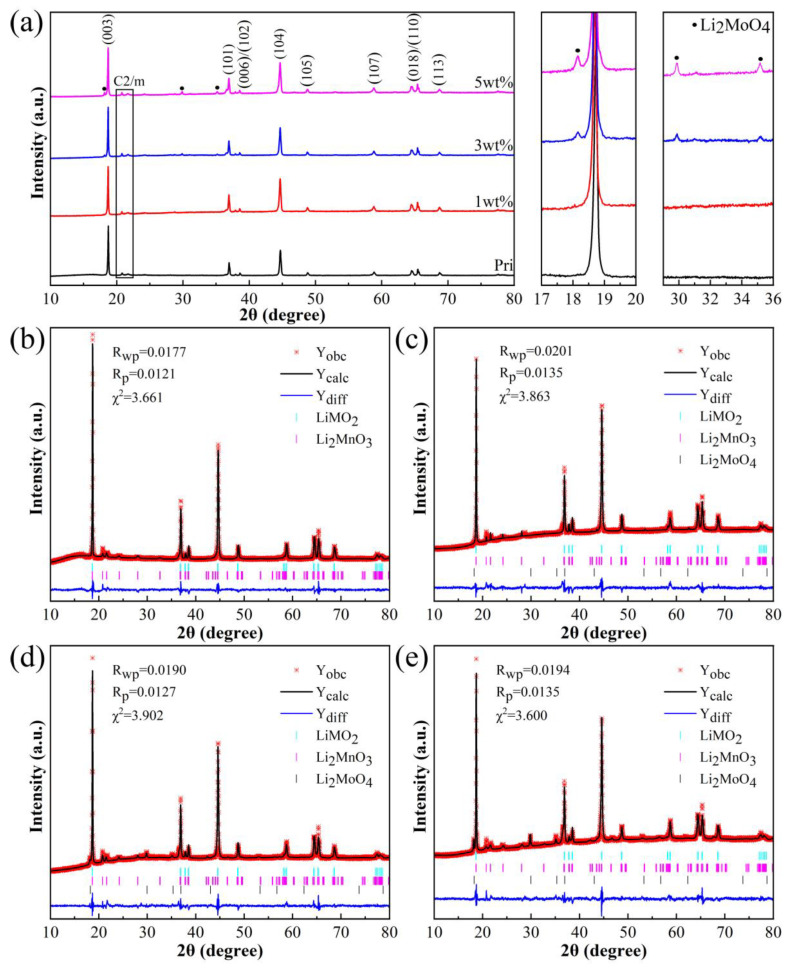
(**a**) XRD patterns and enlarged XRD patterns in selected 2θ range of four samples. (**b**–**e**) Rietveld refinement of XRD patterns (The red lines represent the observed value (measured value), the black lines represent the calculated value, and the blue lines represent the difference between the observed value and the calculated value).

**Figure 2 materials-16-05655-f002:**
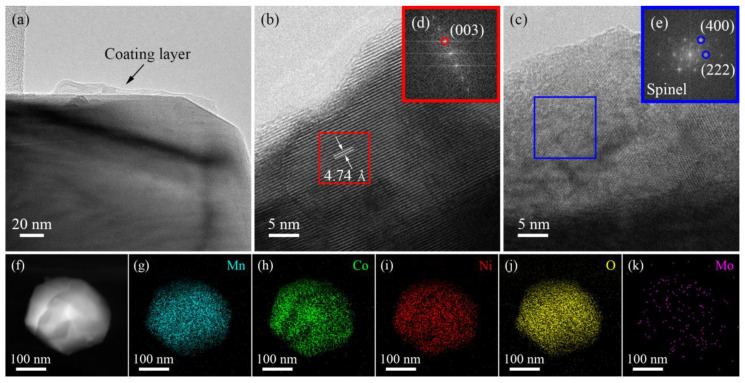
(**a**) TEM image of 3 wt% sample, (**b**,**c**) TEM images of the internal layered structure and the surface spinel structure coating layer of the 3 wt% sample, respectively, and (**d**,**e**) FFT images of the red area in (**b**) and the blue area in (**c**), respectively. (**f**) TEM image of 3 wt% sample, and (**g**–**k**) EDS mapping of Mn, Co, Ni, O, and Mo.

**Figure 3 materials-16-05655-f003:**
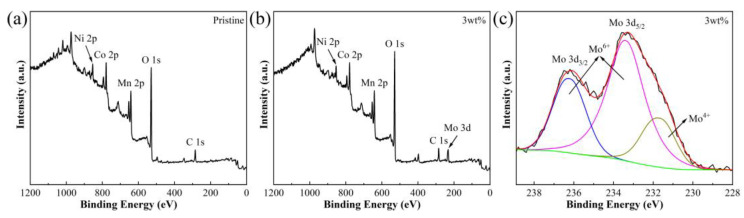
(**a**,**b**) XPS full spectrum for Pristine and 3 wt% samples, and (**c**) XPS spectra of Mo 3d for 3 wt% sample (The black line is the measured value, the green line is the background value, and the lines of other colors are the fitting values).

**Figure 4 materials-16-05655-f004:**
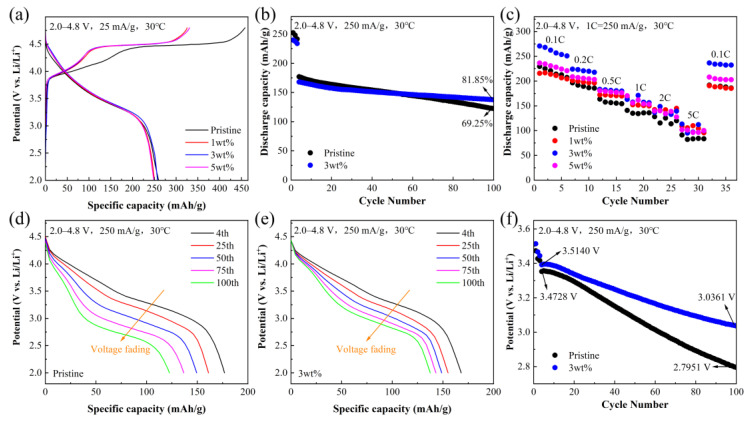
(**a**) Initial charge–discharge curves of four samples at 0.1 C. (**b**) Cycling performance for Pristine and 3 wt% samples. (**c**) Rate performance of four samples. (**d**,**e**) Attenuation of discharge voltage for Pristine and 3 wt% samples, and (**f**) discharge medium voltage attenuation curves for Pristine and 3 wt% samples.

**Figure 5 materials-16-05655-f005:**
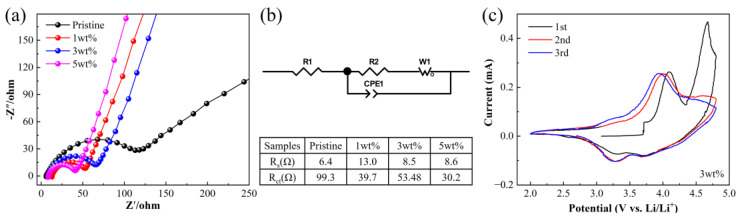
(**a**) EIS of four samples, and (**b**) equivalent circuit diagram and impedance parameters of the samples. (**c**) CV curves of 3 wt% sample.

**Table 1 materials-16-05655-t001:** Lattice parameters of samples from Rietveld refinement.

Samples	I_(003)_/I_(104)_	c	a	c/a	Ni^2+^ in Li Layer (%)	Li_2_MoO_4_ Content (wt%)
Pristine	1.692	14.245782	2.854563	4.990529899	3.83	0
1 wt%	1.258	14.257465	2.855348	4.993249509	5.36	-
3 wt%	1.526	14.248178	2.854860	4.990849989	3.88	3.0158
5 wt%	1.332	14.262967	2.856606	4.992976630	4.30	4.7180

## Data Availability

Not applicable.

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
