# Peer review of "Improved Electrochemical Performance of Li-Rich Cathode Materials via Spinel Li2MoO4 Coating"

_materials, 2023, doi:10.3390/ma16165655_

Round 1

Reviewer 1 Report

In this work, Li-rich cathode materials Li1.2Mn0.54Co0.13Ni0.13O2 coated with Li2MoO4 with spinel structure was prepared by wet chemistry method and the subsequent calcination process. Li-rich manganese-based cathode materials (LRMs) are considered as one of the most promising cathode materials for the next generation lithium-ion batteries (LIBs) because of their high energy density. The process consists of preparation of the pristine Li1.2Mn0.54Co0.13Ni0.13O2 from the carbonate precursor (Mn0.675Co0.1625Ni0.1625CO3) uniformly ground and mixed with Li2CO3 (an excess of 5% Li2CO3 is added to compensate for lithium loss during the elevated calcination step) in an agate mortar. The as-prepared Li-rich material was mixed with a certain amount of ammonium molybdate and calcined at 720 ℃ for 4h to obtain the Li2MoO4@Li1.2Mn0.54Ni0.13Co0.13O2 material. The samples prepared with 1,3,5 wt% Li2MoO4 coating layer were termed as 1wt%, 3wt% and 5wt%, respectively.

1)     Since only MoO3 was added to the Li1.2Mn0.54Ni0.13Co0.13O2 sample, it is unclear how Li2MoO4 can be formed with a lack of Li in the material?

2)     The authors tested the as-prepared samples and showed that only the sample with 3% of MoO3 has the optimal cycling properties. It is not clear why?

3)     Very small characteristic peak in the range of 20-25° demonstrates the presence of another phase Li2MnO3 with the C2/m space group in the samples. What is its role?

4)     Why “rhomboidal” phase is used instead of “orthorhombic”?

5)     Why do the lattice parameters of the cells with 1-5 wt.% of Li2MoO4 in Table 1 change non-linearly?

6)     Why Mo4+ ions are present in the XPS spectra of Li2MoO4-coated material with Mo6+ ions (Fig. 3)?

7)     “The magnification performance curve also indicates that the sample coated with 3wt% Li2MoO4 has higher capacity at higher magnification”. Please, explain.

8)     “The platform of the sample coated with Li2MoO4 becomes shorter near 4.5 V because the formation of the Li2MoO4 coating layer removes some Li from the material and pre-activates the Li2MnO3 component (line 189)”. Needs an explanation.

9)     It is unclear, why 259.65, 250.29, 257.31 and 248.74 mAh/g of the Pristine, 1wt%, 3wt%, and 5wt% samples, respectively, correspond to Coulombic efficiencies of 56.66, 76.92, 77.79 and 75.16% on line 183. Please, explain “how the Li2MoO4 coating improves the cycling performance of the material with only 69.25% capacity retention rate of the original sample after 100 cycles, while the 3wt% sample still had 81.85% capacity retention rate after 100 cycles” (line 195).

10) It’s unclear: “Some Mo6+ is doped in the lattice interior to convert to Mo4+ to maintain electroneutrality”.

11) There are repetitions of some words in the sentences: p.2 – “We coated a Li2MoO4 coating layer”; p.5 – “a spinel structure coating is formed on the surface of the material coated with Li2MoO4”; line 96: - “then cycled to 100 cycles at 1 C”; line 106: “The results indicate that experimental results…”; line 153: - “spinel structure coating is formed on the surface of the material coated with L2MoO4”; line 173: - “some Mo6+ is doped in the lattice interior to convert to Mn4+ to maintain electroneutrality”; line 183: - it is not clear how the Coulombic efficiency of the samples increases when capacity decreases”; line 188: - “long plateau is related to the oxygen activation of Li2MnO3” – But the amount of Li2MnO3 is so small; line 190 – “This is because the formation of the Li2MoO4 coating layer removes some Li from the material and reactivates the Li2MnO3 component”; line 242 – “Li2MoO4 exhibits a unique spinel structure”?

 Moreover, I was surprised to find the same paper of the authors already printed (10.20944/preprints202307.0029.v1). Some other authors came to the same conclusions in their studies:

1)     ACS Applied Materials and Interfaces (2023) // 10.1021/acsami.1c21182

2)     J. Alloys and Comp. (2016) // 10.1016/j.jallcom.2016.03.003

Please, compare their results with yours.

English is acceptable.

Reviewer 2 Report

Dear Authors,

Fantastic work. The work is good to be accepted with a few minor revisions suggested below.

1. In line 68, I suggest the authors mention the exact amount of ammonium molybdate for the community to be able to reproduce the results and verify this work.

2. Line 117 should be paraphrased to something like, 'The XRD patterns of four samples refined to two-117 phase structure model consisting of rhomboidal R-3m and monoclinic C2/m phases. We do not perform Rietveld and it is not XRD curves, it is XRD patterns.

3. In line 31, I suggest authors refer to the recent exciting and relevant work by Prof. Lekakou. They have demonstrated the high capacity and power density of hybridised electrodes which I believe is very relevant here. The references are C. Lekakou et al., Investigating battery-supercapacitor material hybrid configurations in energy storage device cycling at 0.1 to 10C rate, Journal of Power Sources, 561 (2023). DOI: 10.1016/j.jpowsour.2023.232762.

4. I would just remove the SEM results because there isn't anything meaningful in there. I would always suggest that authors should not just use, and report techniques/results used just to fill the pages.

5. Instead of XPS, I would have preferred LEIS to characterise the surface given that LEIS is much better to look at the outer layers of atoms only. But I understand that the technique is too sophisticated and XPS is okay in this case. But I suggest the authors keep LEIS in their minds in their relevant future work.

6. On EIS, I would like to see the Kramers–Kronig residual analysis for the data validation.

Thank you.

Just a few minor changes needed.

Round 2

Reviewer 1 Report

The authors added two new ones, answered the questions of the reviewer and made some additions to the text. However, it's hard to agree with some of them. For example, when answering the question 4, "rhomboidal" should be replaced by "rhombohedral". I'm happy that after reading some of my questions, the authors were not ready to answer them right away and planned to do it in future. I hope it will be done.

The English language needs a little change.

Author Response

Thanks to the reviewer’s comments. According to the reviewer's comments, supplementary explanations were provided in the introduction, including the structural characteristics, high capacity sources, and charge-discharge mechanisms of Li-rich manganese-based cathode materials. And the reasons why surface protection is important are further explained. The word ''rhomboidal'' in the text has been corrected to ''rhombohedral''. In addition, the accuracy of the professional terms appearing in the text has been determined by referring to authoritative literature. Thanks again to the reviewer's comments. There are indeed some shortcomings in this work, and we expect to find the answers to the questions through more in-depth research in the future.